# Electrochemical Microneedles: Innovative Instruments in Health Care

**DOI:** 10.3390/bios12100801

**Published:** 2022-09-28

**Authors:** Zhijun Liao, Qian Zhou, Bingbing Gao

**Affiliations:** School of Pharmaceutical Sciences, Nanjing Tech University, Nanjing 211816, China

**Keywords:** electrochemical, real-time monitoring, biosensor, fabrication method, microneedle

## Abstract

As a significant part of drug therapy, the mode of drug transport has attracted worldwide attention. Efficient drug delivery methods not only markedly improve the drug absorption rate, but also reduce the risk of infection. Recently, microneedles have combined the advantages of subcutaneous injection administration and transdermal patch administration, which is not only painless, but also has high drug absorption efficiency. In addition, microneedle-based electrochemical sensors have unique capabilities for continuous health state monitoring, playing a crucial role in the real-time monitoring of various patient physiological indicators. Therefore, they are commonly applied in both laboratories and hospitals. There are a variety of reports regarding electrochemical microneedles; however, the comprehensive introduction of new electrochemical microneedles is still rare. Herein, significant work on electrochemical microneedles over the past two years is summarized, and the main challenges faced by electrochemical microneedles and future development directions are proposed.

## 1. Introduction

Electrochemistry is a science that studies the charged interface phenomenon formed by two kinds of conductors and its changes [1]. Due to its high sensitivity [2], ease of use, rapid response, economy, and mobile advantages, electrochemistry is widely used in wearable electronic equipment [3], clinical application of activated coating time (ACT) [4], sewage treatment, etc. At the same time, it also has some shortcomings, including the need for large-scale instruments, inconvenience, poor long-term sensitivity, and physiological indicators that often need to be tested in vitro. In addition, Heubner et al. [5] wrote that there were some shortcomings in the use of a 2-electrode half-cell system for the testing of electrochemical performance (such as the continuous decline in the potential stability of the working electrode and cross-talk phenomena). Thus, it is often used in combination with other techniques, such as liquid chromatography, microneedles, and photochemistry.

Microneedle administration can prevent the liver first-pass effect and enzyme reaction damage in the gastrointestinal tract during the oral administration of drugs, thereby doubling drug absorption and utilization rates [6]. The action mode is to punch out a large number of fine pipelines by stimulating the skin and then directly injecting the drugs into the dermis layer and deep subcutaneous tissue so that the active ingredients can penetrate into the skin. According to the difference in drug release mechanisms [7], microneedles can be divided into five categories: solid microneedles [8], hollow microneedles [9], coated microneedles [10], soluble microneedles, and hydrogel microneedles [11]. Each type has different advantageous features, e.g., hollow microneedles can be monitored in real time using built-in electrodes, and soluble microneedles can be rapidly degraded by skin tissue fluid after penetrating into the skin, thus releasing the drugs from the needle body. Moreover, microneedles have the following advantages: painless, noninvasive, macromolecular penetration into the stratum corneum [12], controllable dose, and high bioavailability [13]. Hence, they have been widely applied in drug delivery, beauty and skin care, and disease monitoring (such as the monitoring of glucose and pH content) [14]. In addition, the microneedle-based electrochemical device is expected to solve the problems inherent in electrochemical method detection.

An electrochemical sensor is an apparatus that converts the chemical quantity of a sample and into an electrical quantity based on the electrochemical property of the sample and to perform sensing detection [15]. Among these, miniaturized electrochemical sensors can be integrated in many types of wearable substrates; for example, electrochemical sensors integrated on microneedle patches can reach the dermal matrix fluid of human and animal skin for the systemic transdermal detection of various analytes [16]. In the wake of the development of microelectromechanical system (MEMS) technology [17], many miniaturized electrochemical sensors combined with microneedles have produced a variety of electrochemical microneedle sensors for monitoring biomarkers. Polsky’s team [18] reported a microfluidic chip that used hollow microneedles to extract tissue fluid and drive it into traditional mobile cells for the final potential detection of potassium. However, this method has some possible shortcomings, including involving a complex extraction mechanism and measurement delay. In contrast, electrochemical microneedle sensors can primarily surmount this issue by real-time monitoring [19]. Lu’s research group [20] developed an electrochemical biosensor for the real-time monitoring of glucose content based on microneedles. At 37 °C, the microneedles showed high stability in long-term monitoring and storage, and they were simple to operate when in use, which effectively solved the crux of measurement delay [21]. It has capacious application prospects in wearable and minimally invasive real-time blood glucose monitoring [22]. Electrochemical technology based on microneedles not only achieved fruitful results in monitoring and detection, but also performed well in terms of in drug delivery. A multimicrochannel microneedle microporous platform was proposed by Chang’s team, [23] which cannot only accelerate the movement of drugs to deep tissues [24] but also trigger cell electroporation, thereby enhancing transport across cell membranes and realizing efficient [25], safe and symmetrical intracellular drug delivery for use as a general delivery tool for emerging pharmaceuticals in vivo [26].

Through electrochemical microneedle technology, human physiological indexes can be transmitted to intelligent devices in real time [27], making the prevention and treatment of many diseases feasible and rendering medicine more intelligent and efficient. In addition, it also creates new research directions for drug delivery, real-time monitoring, wearable electronic devices, and other fields. In addition, although there have been many research reports on electrochemical microneedles [28], there is still a lack of comprehensive introduction. In this review, we have provided a comprehensive overview of this field and summarized some of the work of electrochemical biosensors over the past two years [29], including the preparation methods and materials of electrochemical microneedles, their application in the medical field, the monitoring of biomarkers, and future development directions [30], as shown in Figure 1.

## 2. Preparation of Microneedles and Microneedle Electrodes

Electrochemical microneedles are frequently applied to medical detection due to their good sensitivity. However, due to the limitations of the preparation process and materials [31], they are unable to be applied in the market on a large scale. Familiar microneedle preparation technologies include 3D printing technology, magnetic resonance (MR) stretch lithography technology, and two-step soft lithography technology. Each preparation strategy has its own advantages and disadvantages. The manufacturing cost of 3D printing technology and magnetorheological stretching photoetching technology is low, and the operation method is simple. In addition, the latter does not require masks and light irradiation, and simultaneously, does not need require an adjustment to the stretching temperature. However, the resolution of the microneedle electrodes prepared by the two methods is lower. The two-step soft lithography technology and the two-photon polymerization technology can provide high resolution and repeatability, but possess high manufacturing cost and complex steps, and the maximum manufacturability height of the microneedle array prepared by the two-step soft lithography technology is limited. As an important part of the preparation process of electrochemical microneedles, the preparation of microneedle electrodes has been rapidly developing along with the progress of microelectromechanical technology [32], and an increasing number of preparation processes and materials are emerging. Common microneedle electrode preparation materials are divided into three categories: polymers, metals, and silicon. The following summarizes several current commonly used microneedles and microneedle electrode preparation methods.

### 2.1. 3D Printing

The use of 3D printing technology can rapidly realize random shapes. Toy models, mechanical components, tablets, human tissues [33], human organs, etc., can be printed out through this technology. The common printing process is to use software to design the printed objects, divide them into a series of digital slices, and then transfer the information of these slices to the 3D printer. Ultimately, the thin layers can be continuously stacked sourcing a special mucilage using the 3D printer until a solid object is formed. Since this technology has the advantages of low cost and simple processing, it has been extensively used in architectural design, manufacturing, the food industry, and other fields. Common printing materials are resin, hydrogel, plastic, and metal powder. Currently, 3D printing technology has steadily grown into a frequently used method for preparing microneedles. For example, Xu et al. [34] prepared a microneedle (Figure 2a) using photopolymerization 3D printing technology with grooves on the facade that allow liquid to flow from the needle tip to the base under the influence of capillary forces, and the resulting microneedle device can penetrate the skin and tissue and collect liquid samples in a skin model. Although 3D printing technology can be used to design the microneedle geometry, spacing, height, and number, this technology is limited by the real resolution, which can affect the penetration and perception of microtargeting to the skin. With the development of this technology, two-photon polymerization also appeared in the field of vision. The advantage of this new 3D printing technology is that it can prepare high-resolution structures. However, the high price limits its application in the preparation of microneedles.

### 2.2. Magnetorheological Drawing Lithography

Most microneedle manufacturing methods exhibit the problems of high manufacturing cost, long time consumption, intricate steps, and expensive equipment, but few researchers have addressed the optimization of the process when studying microneedles. The superiority of microneedles, which are painless and minimally invasive, support their broad used in the medical field. Therefore, Jiang’s team [35] used magnetorheological drawing lithography (MRDL) to manufacture microneedles (Figure 2b). They used cylindrical Nd-Fe-B permanent magnets to generate an external magnetic field and selected a copper pin with a diameter of 0.7 mm as the pull column. Thirty nanoliters of hemispherical curable magnetorheological fluid (CMRF) droplets were immersed and suspended at the top of the column (resins and magnetic powders are ordinarily used to fabricate CMRFs). The droplets were then driven toward the substrate and compressed against the upper surface of the substrate for 300 ms. Then, the liquid microneedles were deposited on the substrate at 1 mm/s under the action of an external magnetic field. Finally, the liquid microneedles were placed in an external magnetic field and prebaked by hot air blowing at 95 °C for 5 min, and the prebaked microneedles were cured in an oven at 100 °C for 1 h. The method not only inherits the virtues of the thermal stretching method without a mask or light irradiation [40], but also eliminates the requirement of adjusting the stretching temperature. In addition, the low manufacturing cost and the simple and easy manufacturing method are favorable for MN manufacturing to move toward batch and industrial production.

### 2.3. Two-Step Soft Lithography

Photolithographic technology is a process that uses the principle of photochemical reactions and chemical and physical etching methods to transfer the pattern on the mask plate to the wafer. Its principle originates from photoengraving in printing technology [41]. Currently, as a mature technology, it is at the forefront of the manufacture of silicon microneedle arrays. In addition, it is extensively used in the manufacture of hollow, solid, and planar microneedle arrays, providing high repeatability and high resolution for the manufacture of solid and hollow microneedles. Xu et al. [36] developed a representative conductive microneedle array (Figure 2c) using a two-step soft lithography technique with hyaluronic acid, poly (3,4-ethylenedioxythiophene)/poly (styrenesulfonate) (PEDOT: PSS), manganese, and stainless steel. For painless dental anesthesia, the team used conductive microneedles to accelerate ion introduction and target the delivery of anesthetics to bone tissue, greatly improving bioavailability. However, this technique also faces some challenges, such as the numerous steps involved, the high manufacturing cost, and the fact that the maximum manufacturability height of the microneedle array is still determined by the thickness of the silicon wafer used.

### 2.4. Plating

Along with the rapid development of MEMS technology, an increasing number of researchers have applied electrochemistry to microneedles and prepared microneedle-based electrochemical sensors. Among these technologies, microneedle electrodes are an important part of electrochemical microneedles. The most familiar preparation method is to modify the bare microneedles with different coatings (the common coatings include carbon coating, Ag/AgCl layer and polyurethane). A typical two-electrode electrochemical microneedle possesses a working electrode (WE) and an RE. For example, Crespo et al. [37] prepared a two-electrode electrochemical microneedle (Figure 2d), which was modified with different coatings and polymeric films, and the potassium selection electrode and reference electrode for potential reading were prepared. Moreover, in vitro experiments proved that this method had sufficient selectivity and good repeatability and was suitable for the clinical detection of potassium in ISF (skin interstitial fluid). Another common example is the three-electrode microneedle, which has one more counter electrode than the two-electrode microneedle. As shown in Figure 2e, Zhang’s group [38] developed a three-electrode microneedle, installing the electrode after plating in the opening of the hollow microneedles so that the electrode was in direct contact with the skin interstitial fluid, and the fluctuation of the electrolyte concentration was monitored in real time. In addition, Voelcker’s team [39] also prepared a three-electrode electrochemical microneedle, as shown in Figure 2f, and applied the prepared high-density microneedle sensing patch for transdermal blood glucose monitoring. The sensing patch can perform minimally invasive in situ glucose detection in ISF without extracting the ISF. Although the plating layer has a relatively mature fabrication process, the prepared patch can only be used for detecting glucose or some other biomarker, and the metal used for the plating layer is frequently a precious metal, such as gold or silver. Hence, in the future, scientific researchers may begin manufacturing a microneedle capable of detecting multiple biomarkers in ISF, amplifying the application range of the microneedle and enhancing its practicability.

## 3. Monitoring Physiological Indicators

With the rapid expansion of electrochemical technology, the level of medical detection has been prodigiously enhanced. Researchers have a passion for building a monitoring device with a long monitoring time, ease of use, high sensitivity, and benign biocompatibility [42]. Microneedle-based electrochemical sensors have become a popular research area due to their high sensitivity and straightforward application [43]. There are many secretions with biomarker properties in demic ISF. Some physiological and biochemical indicators in ISF can be extracted and monitored by electrochemical microneedle sensors, which can simplify the monitoring steps and curtail the monitoring time, thus guiding doctors to make diagnoses and treatments in a timely manner.

### 3.1. Multiple Monitoring for Na^+^ and K^+^

Electrolytes play a significant role in regulating muscle activation, cardiovascular function, and hydration. Their disfunction may bring about a series of diseases (such as hypokalemia, hyponatremia, hyperkalemia, and hypernatremia), causing damage to the organism. Therefore, in terms of clinical application, to reasonably control the patient’s condition, doctors monitor the electrolyte concentration to determine the patient’s health. At present, the adopted electrolyte monitoring method is generally completed through blood sampling, and in the process of blood collection, subjects will experience discomfort, e.g., pain, and the fear of needles for some subjects, especially children, aggravates the difficulty of blood collection. As shown in Figure 3a, Narayan’s research group [44] fabricated an all-solid-state ion sensing device, in which the screen-printed electrode is composed of an all-solid reference electrode and a working electrode. During the potential measurement operation, the interior of this solid-state microneedle sensor does not need to be filled with solution (used to achieve a constant dipole potential), providing an innovation for the design of potentiometer sensors. Based on the similar electrolyte components of ISF and plasma, Zhang et al. [38] also developed a microneedle potential sensing system for multichannel monitoring of Na^+^ and K^+^ for the extraction of ISF. In addition, as shown in Figure 3b, the sensor system reported by them has the advantage of rapid response time, good reversibility and repeatability, and appropriate selectivity in the measurement of Na^+^ and K^+^, and has prospects for broad applications in the fields of medical health and real-time monitoring.

### 3.2. Monitoring pH

There are two sources of acid-base substances in the human body. One is produced by normal metabolism of the body, and the other is ingested through daily diet. With the changes in acid–base substances in the human body, the body realizes the balance state through continuous regulation. It has been proven that acid–base properties are closely related to many diseases, including atherosclerosis, hypertension, fatty liver disease, and senile osteoporosis. Therefore, the real-time monitoring of pH changes has high practical value in clinical practice. Against this background, many scientific research teams have also developed pH sensors that can monitor the hearts of rats, as well as the cerebrospinal fluid and bladders of mice, but none of them has solved the problem that the sensing elements may fall off when the microneedles are used percutaneously. As shown in Figure 3c, Zhang et al. [45] created a high-performance electrochemical pH microneedle for the real-time monitoring of changes in pH in the rat brain, which not only has capital selectivity for other potential interfering substances in the brain, but also exhibits excellent responsiveness to changes in pH in vivo. In addition, the new potentiometric method provided by this work can monitor the dynamic pH change in vivo with high reliability and stability. Meanwhile, as shown in Figure 3d, Lemieux’s research group [46] also reported a capacitive electrochemical microneedle sensor with excellent selectivity and sensitivity which can simultaneously measure pH, nitrate, and phosphate. The pH microneedle sensor responded within the pH range of 3–10. It plays a significant role in soil and water quality analysis, as well as agriculture and food quality assessment.

### 3.3. Monitoring of Ketone Bodies

Ketoacidosis is a complication of diabetes, which is mainly due to severe metabolic disorder syndrome caused by insufficient insulin secretion and excessive hormone secretion which antagonizes insulin. In this process, the human body accumulates an excessive amount of ketone, a harmful acidic substance. Hence, to better manage diabetes mellitus [51], a method of monitoring ketone bodies is urgently needed. Although there are many apparatuses for monitoring the content of blood glucose, and prodigious progress has been made in the management of diabetes, the problem of continuous monitoring of ketone bodies, which can lead to hyperglycemia and metabolic acidosis, has not been solved. Based on this, as shown in Figure 3e, Wang et al. [47] manufactured a microneedle platform for the continuous monitoring of ketone bodies using β-hydroxybutyrate (HB) as the main marker of ketone formation and the β-hydroxybutyrate dehydrogenase reaction as the basis. Hence, the developed platform shows high selectivity, high sensitivity, and good stability, with broad application prospects for the minimally invasive monitoring of ketone bodies and great application potential for the precise management of blood glucose.

### 3.4. Determination of Abscisic Acid

Abscisic acid is a plant hormone that can causes bud dormancy and leaf shedding, inhibiting cell growth. Gibberellin can stimulate leaf and bud growth and increase yield. Hence, several effects of abscisic acid can be counteracted by gibberellin. For example, gibberellin can be used to surmount the elongation of genetically high-stalk corn and the inhibition of seed and potato germination. Abscisic acid is widely distributed in higher plants, and its content is changed under the influence of the plant growth environment [52], planting method, and growth cycle. It plays a significant role in plant physiology. Therefore, to research the growth mechanism of plants and further regulate their growth, researchers have developed an enormous interest in the determination of trace abscisic acid, especially the content of abscisic acid in living plants. Several measurement methods have been developed, including high-performance liquid chromatography and liquid chromatography-mass spectrometry. However, most of these techniques require the purification of abscisic acid from plant samples, which is time consuming. Moreover, these methods cannot monitor the level of abscisic acid in living plants in real time. Therefore, Li et al. [48] developed a sensor using SnO_2_ as the gold catalyst carrier and a direct current (DC) arc plasma jet chemical vapor deposition (CVD) method to grow graphene vertically on tantalum wires, with a diameter of 0.6 mm. Then, the graphene microelectrode, a Ti line, and a Pt line are packaged in a microneedle array sensor with a three-electrode system, and abscisic acid is quantitatively detected through direct electrocatalytic oxidation. The detection mechanism of ABA in plants is shown in Figure 3f. In addition, the electrochemical sensor can monitor abscisic acid in living plants online, showing a wide pH adaptation range and abscisic acid response concentration range, high sensitivity, a low detection limit and good long-term stability, and exhibits great application potential in the field of agricultural production.

### 3.5. Monitoring Glucose

Diabetes is ordinarily diagnosed by measuring the blood glucose level, which reaches the stipulated range for the initial diagnosis of diabetes when the fasting blood glucose level of the human body (without any calorie intake for at least 8 h to 12 h) is greater than 7.0 mmol/L and the 2 h postprandial blood glucose level is greater than 11.1 mmol/L. A common blood glucose measurement method is to use a lancet to remove the blood from the fingertip in an invasive manner, placing the blood in the glucose meter for detection. This method has the superiority of simple operation and low cost, but it has the potential for infection, and it causes pain. Thus, as shown in Figure 3g, Stefano’s research group [49] manufactured a biosensor for monitoring glucose by performing standard photolithography on polyethylene glycol diacrylate doped with an enzyme, a redox mediator, and a photoinitiator. Compared with the conventional blood glucose monitoring method, the microneedle-based biosensor has the advantage of being painless and minimally invasive, as well as greatly reducing the risk of infection. In addition, Cho et al. [50] also reported an electrochemical sensor for monitoring glucose, which used the nonenzymatic (direct) detection of glucose with a linear response within the physiologically relevant range (1–40 mmol/L), with high sensitivity, as well as good repeatability and stability, as shown in Figure 3h. In addition, the sensor can selectively detect glucose, even when the concentrations of ascorbic acid, lactic acid, and dopamine are 10 times higher than normal.

### 3.6. Wearable Biochemical and Physiological Sensors

With the growth of society, people have paid more attention to health monitoring, and there has been an increase in electrochemical technology; thus, researchers have made begun creating wearable biological devices to monitor health. In addition, wearable biological devices have the advantage of the dynamic, continuous, and real-time monitoring of human physiological information. Microneedle-based biosensors, as wearable devices, have prodigious value. They are superior in that they can measure physiological indicators in body fluids by noninvasive means, reflecting them on intelligent devices. Such biomarkers generally include saliva, interstitial fluid, tears, and metabolites in body fluids. For example, apomorphine (APO) is currently commonly used in the treatment of patients with PD (Parkinson’s disease) to minimize PD fluctuation due to irregular high-dose medication. As shown in Figure 4a, Wang et al. [53] developed a wearable electrochemical sensing platform based on microneedles for the continuous monitoring of APOs in artificial ISFs. The microneedle APO sensor has extremely high sensitivity and a wide measurement range (covering clinically relevant concentrations) and exhibits favorable stability and high selectivity for APO, making the developed sensor platform a candidate product for wearable minimally invasive monitoring. As shown in Figure 4b, Crespo’s team [54] also prepared a wearable sensing device coupling the newly developed microneedle patch with a portable potentiometer to measure the percutaneous pH of euthanized rats. At the same time, the subcutaneous pH measurement and the extraction and analysis of ISF samples were evaluated, and the former was more reliable than the latter. From extraction, to in vitro analysis, to human measurement, the research and application of wearable electronic devices based on microneedles have made effective progress.

### 3.7. Integrated Chips

The superiority of high performance, miniaturized, low-power consumption integrated chips allows for their extensive use in aerospace, medical health, computers, satellite navigation, intelligent robots, and other fields. Along with the rapid expansion of microprocessing technology, integrated chips are constantly being upgraded. For example, as shown in Figure 4c, Moschou et al. [55] prepared a printed circuit board-integrated chip for glucose detection with high sensitivity, high specificity, and good reproducibility, paving the way for the first cost-effective, indolent diabetes management microsystem. As shown in Figure 4d, another three-electrode system for glucose monitoring was exploited by Voelcker’s team [39], which realized the application of a high-density microneedle sensing patch for transdermal blood glucose monitoring. The microneedle sensing patch can perform minimally invasive in situ glucose detection from ISF without extraction or collection, and more importantly, the painless and minimally invasive detection method provides a new detection method for existing transdermal diagnosis.

## 4. Drug Release

Drug release refers to the fact that the system delivers the drug to the patient at a specific speed in an expected way so that the stable dosing speed and high bioavailability can be maintained for an extended time. The decomposition capability of digestive juices such as saliva, gastric juices, and intestinal juices greatly diminishes the efficacy of many oral medications and reduces bioavailability [56]. In addition, drugs may also be converted in the gastrointestinal tract to decrease efficacy and even engender toxic effects. Therefore, how to reasonably and safely convey and release drugs and finding new drug delivery methods are difficult problems which have become hot topics in medical treatment research [57].

### 4.1. Release of Insulin

Diabetes is a familiar metabolic disease that has threatened hundreds of millions of people worldwide; it can also trigger chronic damage to the heart, blood vessels, kidneys, nerves, and other tissues and organs. There are many causes for diabetes, the most common being genetic factors, inadequate exercise, excessive eating, and other environmental factors. The traditional treatment is the manual injection of insulin to regulate blood glucose; patients experience discomfort, such as pain, and there is a risk of infection. Hence, to address these issues, Xie et al. [58] exploited a fully integrated closed-loop system (Figure 5a) based on a microneedle platform and wearable electronics that could regulate the release of insulin by loading insulin concentration, iontophoresis current, and duration. Insulin could also be flexibly added to the device without being limited by the loading dose. Therefore, microneedle platforms can deliver insulin in a way that can prevent pain and potential infection.

### 4.2. Drug Release to Bone Tissue

Since the advancement of medicine and the maturation of local anesthesia technology, anesthesia has become the main means of modern dental treatment for toothache. The common anesthesia method uses a needle and suction syringe to inject the anesthetic into the oral submucosa close to the root for anesthesia. Then, the drug is allowed to randomly diffuse into the alveolar bone and reach the nerve endings of the teeth. However, this type of administration can give rise to anxiety in patients prior to dental treatment, since the needle will penetrate into the mucosal tissue and the patient will experience stress as the anesthetic solution diffuses. Thus, Xu et al. [36] exploited a conductive microneedle for the targeted delivery of anesthetic to bone tissue (Figure 5b). The solid model of a rabbit mandible slice and two pig skins verified that the designed conductive microneedles and iontophoresis (ITP) platform could enhance the permeability of small drug molecules through soft and hard tissues. The advantages are that the drug administration speed is accelerated with the increase in drug penetration depth, and the excellence of noninvasive and minimal pain greatly relieves the anxiety of patients in regards to seeing a doctor. Hence, the minimally invasive transdermal nature of this conductive microneedle makes it a viable replacement for hypodermic needles and demonstrates efficacy in transdermal drug delivery applications.

### 4.3. Delivery of Adriamycin

Cancer, as a general term for a series of malignant tumors, has a high mortality rate, and its symptoms are often noted in the late stage. Its prevention and treatment are hot topics. Cancer cells are cells whose genome has mutated, and they have escaped and survived the clearance of the immune system. They have the ability to breed and spread indefinitely. By seizing the growth resources of healthy cells, they inhibit the growth of normal cells, thus endangering human health. The idiographic etiology of cancer is still unclear and may be related to heredity and long-term exposure to radiation. Some behaviors increase the risk of cancer, including alcohol abuse, smoking, and long-term overeating. In addition, the most common treatments available today are surgery, radiation therapy, chemotherapy, molecular targeted therapy, and immune checkpoint inhibitor therapy [60]. Adriamycin is a chemotherapy drug that acts on the chemical structure of nucleic acids. It is effective for both acute lymphoblastic leukemia and myeloid leukemia and has a corking antitumor effect. However, it is also highly cytotoxic. Thus, the dosage of Adriamycin (doxorubicin) should be strictly controlled in the clinic to avoid causing more harm to the body. Based on this, as shown in Figure 5c, a microquantitative platform with multichannel microneedles, which can efficiently, safely, and uniformly deliver Adriamycin to cells in vivo, was exploited by Chang’s team [23]. The microneedles were prepared by high-precision 3D printing, providing a concentrated and safe electric field, which not only accelerated the speed of Adriamycin entering deep tissues under electrophoresis, but also triggered cell electroporation, enhanced the trans-cell membrane transport, reduced cytotoxicity, and had significant anticancer effects. Hence, it has broad application prospects in the field of drug delivery.

### 4.4. Delivery of Glucocorticoids

Atopic dermatitis is a chronic inflammatory skin disease that is accompanied by problems such as dry skin and severe pruritus, seriously affecting patients’ daily lives. In the clinic, it is treated with topical glucocorticoids. However, due to the barrier effect of the skin, the probability of drug molecules being used is very low. Therefore, it is particularly significant to develop an effective drug transportation method. Guo’s group [59] reported a two-electrode microneedle patch (Figure 5d), composed of a polylactic acid-platinum microneedle array and a polylactic acid-platinum-polypyrrole microneedle array, whose drug release rate can be influenced by changing the voltage, or the drug loading rate can be influenced by changing the polymerization time and drug concentration. The controlled release system based on the dual-electrode microneedle patch only releases drugs under the condition of electric stimulation, and rarely releases drugs under the condition of no electric stimulation. In addition, the developed intelligent patch has copacetic mechanical properties, excellent electrochemical properties, and exceptional drug loading capacity, giving it wide application prospects in the clinical treatment of atopic dermatitis, with promising application potential in the direction of a closed-loop intelligent drug delivery system [61].

### 4.5. Summary

With the progress and expansion of society and technology, the living standards of human beings have gradually ameliorated. Research is no longer limited to the pursuit of medical health, but it also focuses on the improvement of medical comfort, which establishes higher requirements for the medical field. The new platforms designed for the release of anesthetics in this study offer significant potential for rapid dosing, improved efficacy, and a reduction in the number of administered doses. The microneedle platform used in releasing insulin is combined with a real-time monitoring system to form a closed-loop treatment system. The system for the timely administration of medicine to patients is convenient and easy to operate, and patients can self-administer medicine at home, which is of great significance for many areas with backward medical treatment conditions [62]. As a multifunctional platform, microneedles can be used for drug delivery and transdermal diagnosis. Hence, many researchers have focused on the practical application of electrochemical microneedles in medicine-drug delivery. In this aspect, conductive microneedles offer exceptional use characteristics, including biological safety, therapeutic effects, ease of self-administration, and painless skin penetration [63], making them applicable to high-precision treatment as a substitute for traditional needle administration.

## 5. Conclusions and Perspectives

In this review, the development and application over the past two years of electrochemical sensors based on microneedles are introduced, and the preparation methods, integrated chips, drug delivery methods, material of microneedles (Table 1), and the monitoring and detection of biomarkers (including real-time monitoring of subcutaneous potassium content and blood glucose concentration) are summarized. Although electrochemical microneedle technology has developed rapidly over the years, there are still some limitations and challenges, including the fact that microneedles are easily broken due to their large dimensions, their inability to sufficiently release drugs, the difficulty of stably inserting them into the skin for an extended period, high manufacturing costs, the insufficient biocompatibility of materials, inadequate drug loading, and their insufficient mechanical strength. In addition, the environment often interferes with its repeatability and stability due to its high sensitivity. Therefore, the research still a long way to go to address these challenges. Some new technologies have begun solving these problems. For example, the stability and durability of the chip have been greatly improved; thus, the application of the chip in the sensor is expected to improve repeatability and stability. To achieve greater breakthroughs, researchers need to further explore this topic. Several perspectives of electrochemical microneedles presented in the following.

Microneedle-based electrochemical sensors have shown great potential. However, since electrochemical sensors depend on the detection of electrical signals, transmitting electrons to the signal receiving end of the sensor is also a challenge. At present, electrochemical sensors often require additional equipment to detect and analyze signals, and electrons are transmitted based on wires, which greatly reduces the usability of wearable sensors. With the popularization of short-distance high-frequency wireless communication (NFC), wireless charging, Bluetooth, and other technologies, compared with the traditional electronic transport mode, wireless transmission can be applied to electrochemical sensors for preparing portable electrochemical devices, which can solve the problem of electronic transmission, to a large extent, and reduce the size of the device.

A growing number of researchers are devoted to the development of electrochemistry. However, the preparation of electrochemical microneedle electrodes often requires precious metals, such as gold and silver, which greatly increases the preparation cost. Therefore, it is possible to try to develop lower-cost metals for preparing electrodes, such as iron and copper. However, complex traditional processing technology, including coating and electrodeposition, will increase the manufacturing cost and reduce the commercial value. Emerging 3D printing technology can reduce their cost, and artificial intelligence technology can provide a better method for the analysis of microneedle data, thus increasing its practicality.

In recent years, microneedles have developed rapidly and have been applied in many fields. However, it is difficult to realize large-scale production, which restricts the application of microneedles. Therefore, we hope that enterprises that study microneedles in China and abroad will invest more energy in the optimization of the preparation process, achieving preparation technology with lower cost and higher resolution. At the same time, we also hope that the government will introduce some preferential policies to guide enterprises to allow microneedle technology to benefit the public as soon as possible.

In summary, the microneedle-based electrochemical sensor has become an indispensable technology on the road to modern scientific and technological progress and has been applied to all aspects of life. This review hopes to encourage researchers in different fields to jointly explore the application of electrochemical microneedles and solve the current problems and challenges, attaining greater value and achieving the successful marketing of electrochemical microneedles.

## Figures and Tables

**Figure 1 biosensors-12-00801-f001:**
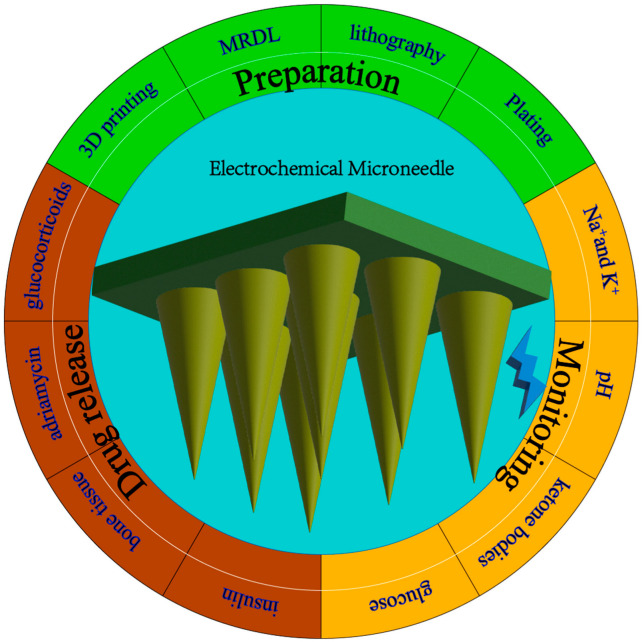
Electrochemical microneedles for healthcare.

**Figure 2 biosensors-12-00801-f002:**
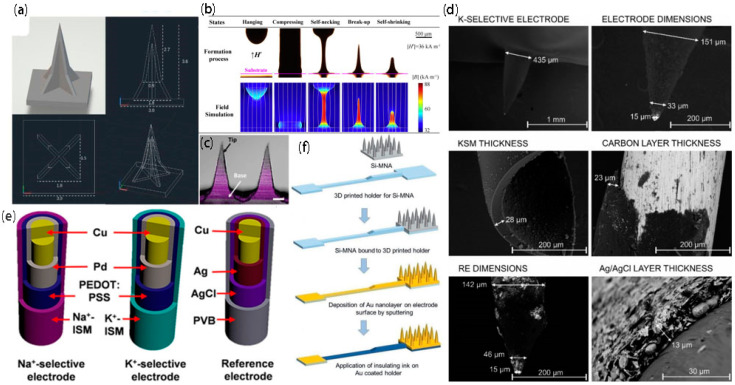
Preparation technology of microneedles. (**a**) Microscopic image of a 3 × 4 MN array [34]; copyright 2021, John Wiley and Sons. (**b**) Formation process of a liquid microneedle with the MRDL method under an external magnetic field of 36 kA m^−1^ [35]; copyright 2017, Elsevier. (**c**) The Cy5 dye shows the cross-sectional profile of the entire MN tip [36]; copyright 2021, Wiley-VCH GmbH. (**d**) SEM images of a microneedle-based potassium-selective electrode, magnification of the tip of the microneedle-based potassium-selective electrode, potassium-selective membrane (KSM) peeled off from the microneedle, cured carbon layer on top of a solid microneedle, microneedle-based reference electrode (RE), and Ag/AgCl layer of the RE [37]; copyright 2018, American Chemical Society. (**e**) Functional layer diagram of the sodium ion-selective electrode, potassium ion-selective electrode, and reference electrode modified by multiple layers [38]; copyright 2021, American Chemical Society. (**f**) Si-MNA substrate attached to the 3D printed holder sputter deposition of a thin film of Au [39]; copyright 2021, Wiley-VCH GmbH.

**Figure 3 biosensors-12-00801-f003:**
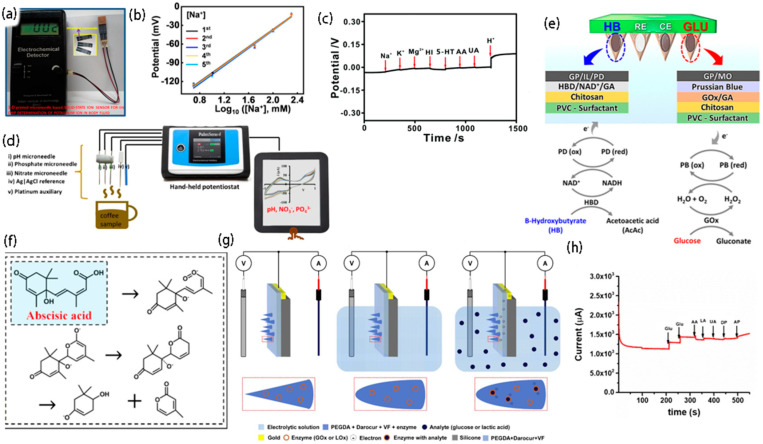
Monitoring of biochemical indicators. (**a**) Microneedle-assembled all−solid−state potassium ion sensor [44]; copyright 2020, Wiley Periodicals LLC. (**b**) The repeatability of the Na+ sensor [38]; copyright 2021, American Chemical Society. (**c**) Selective measurement of MoS_2_/PAN/AN with the addition of a series of interfering substances [45]; copyright 2018, The Royal Society of Chemistry. (**d**) Stainless steel electrochemical capacitance microneedle sensor capable of simultaneously measuring pH value, nitrate, and phosphate [46]; copyright 2021 Springer Nature. (**e**) Schematic representation of dual-marker sensing on a microneedle sensor platform [47], copyright 2020, American Chemical Society. (**f**) Detection mechanism of ABA in plants [48]; copyright 2021 Elsevier. (**g**) Working concept sketch of the three-electrode system [49]; copyright 2016 Elsevier. (**h**) Chronoamperometric response of Au/Pt black/nafion microneedle electrodes with three in the presence of common interfering substances in + 0.12 V buffered saline [50]; copyright 2018, Springer Nature.

**Figure 4 biosensors-12-00801-f004:**
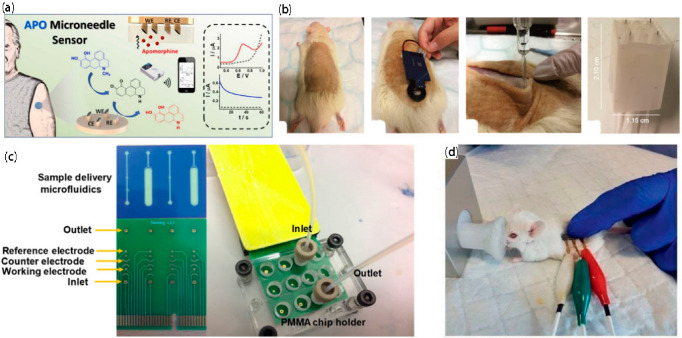
Application of an integrated chip and wearable sensor. (**a**) Usage model of APO sensor [53]; copyright 2021, Elsevier. (**b**) pH patch coupled with the potentiometric electronic board inserted into shaved rats prior to euthanasia to provide transdermal measurements from the rat’s back [54]; copyright 2021, American Chemical Society. (**c**) Illustration of the fully integrated Lab-on-PCB microsystem [55]; copyright 2020, Elsevier. (**d**) Application of the patch on shaved mouse skin [39]; copyright 2021, Wiley-VCH GmbH.

**Figure 5 biosensors-12-00801-f005:**
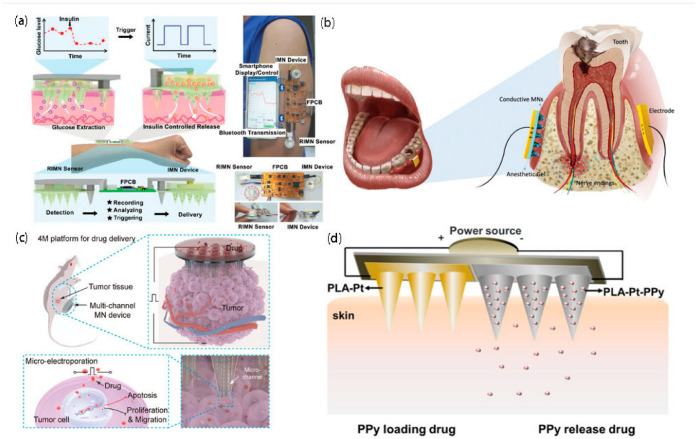
Practical application of drug release. (**a**) MN-based system for real-time and in situ diabetes monitoring and treatment by wirelessly communicating with a smartphone for data transmission and display [58]; copyright 2021, John Wiley and Sons. (**b**) Schematic diagram demonstrating the enhanced effect of conductive MNs on ITP for directed delivery of anesthetic drugs to deep tissues in dentistry [36]; copyright 2021, Wiley-VCH GmbH. (**c**) Schematic diagram of a platform for local administration in vivo. The inset shows an enlarged detail of the platform attached to a solid tumor. An external electric field is applied from the needle tip to the bottom (top) of the drug reservoir of the device. The lower right panel illustrates the drug delivery process at a single MN level. The illustration on the left shows that the drug molecules enter the tumor cells by electroporation, exerting their anticancer effects [23]; copyright 2021, Wiley-VCH GmbH. (**d**) Schematic diagram of drug release from a controlled transdermal drug delivery system by electrical triggering [59]; copyright 2022, American Chemical Society.

**Table 1 biosensors-12-00801-t001:** Materials and shapes of microneedles and their applications in the field of health.

S.No.	WE Material	RE Material	Height (μm)	Base Diameter (μm)	MN Shape	Application	Ref.
1	steel, reduced graphene oxide and Pt nanoparticles, PVP	steel, Ag/AgCl	800	225	circular cone	Detect H_2_O_2_	[64]
2	stainless steel, carbon	stainless steel, Ag/AgCl	-	435	circular cone	Detect potassium	[37]
3	graphite powder, mineral oil	graphite powder, mineral oil, Ag/AgCl	1500	425	triangular pyramid	Monitoring of nerve agents	[65]
4	carbon, Cr, Au	Ag/AgCl	600	400	circular cone	Detect glucose	[58]
5	graphite powder, mineral oil, rhodium nanoparticles	Ag/AgCl	-	500	The cylinder is chamfered by a plane.	Monitoring of apomorphine	[53]
6	manganate, polylactic acid carboxyl multiwalled carbon nanotubes	Ag/AgCl, Polylactic acid carboxyl multiwalled carbon nanotubes	870	250	circular cone	Monitoring electrochemical changes in the skin	[66]
7	stainless steel, HA, PEDOT:PSS, Mn	stainless steel, PEDOT:PSS, HA, Mn	550	300	pyramid	Drug delivery	[36]
8	Au, polystyrene	Ag, Mn	1000	750	circular cone	Monitoring sodium	[67]
9	graphite powder, mineral oil, tyrosinase	Ag/AgCl	1500	425	pyramid	Monitoring levodopa	[68]
10	stainless steel, carbon, Ag/AgCl	stainless steel, carbon, Ag/AgCl	500	400	circular cone	Detect PH	[54]
11	Au, Si	Si, Ag/AgCl	250	50	circular cone	Detect glucose	[39]
12	Pt	Ag/AgCl	700	-	needle tubing	Detect glucose	[69]
13	carbon−polymer composite, manganese	Ag/AgCl	350	200	pyramid	Drug delivery	[70]

## Data Availability

Not applicable.

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
