# Peer review of "Electrochemical Microneedles: Innovative Instruments in Health Care"

_biosensors, 2022, doi:10.3390/bios12100801_

Round 1
Reviewer 1 Report
This manuscript structure is not up to the mark in all aspects eg. Lots of syntax errors, unnecessary words, and application of irreverent words to overcome plagiarism which makes it more complicated. Authors should rewrite from the top to bottom and extensively modify this manuscript. Please see some of the examples below--
1. 3rd sentence in the abstract part is too long and confusion. Authors should simplify it.
2. “market as analytical tool for what reason?’ I think authors should select a different word
3. INTRODUCTION- First paragraph is not completed and not clear for the non-expert, average expert, or new researcher. It should rewrite with a clear conclusion.
4. Figure -1 should be redrawn using for a better looking.
5. The manuscript is written poorly, very hard to understand many sentences. There is no connection between two consecutive sentences. Overall messages are unclear almost in all the paragraphs. Therefore, it should be extensively checked by the native language editor.
Author Response
Response to Reviewer Comments
Reviewer comments in normal type
Author response in italics
Verbatim quotes from the revised text are in boldface
Reviewer1:
This manuscript structure is not up to the mark in all aspects eg. Lots of syntax errors, unnecessary words, and application of irreverent words to overcome plagiarism which makes it more complicated. Authors should rewrite from the top to bottom and extensively modify this manuscript. Please see some of the examples below—
Reply: Thank you very much for your patient review, we have carefully revised the related issues. We redraw figure-1, carefully re-checked and thoroughly revised the manuscript comments, and all modifications are highlighted.
- 3rdsentence in the abstract part is too long and confusion. Authors should simplify it.
Reply: We carefully considered your valuable comments, the problematic sentence has been modified and polished, and all modifications are highlighted.
Recently, microneedles have combined the advantages of subcutaneous injection administration and transdermal patch administration, which is not only painless but also has high drug absorption efficiency.
- “market as analytical tool for what reason?’ I think authors should select a different word
Reply: We have carefully considered your valuable comments, and the words in question have been revised. All modifications are highlighted.
Therefore, they are commonly applied to laboratories and hospitals.
- INTRODUCTION- First paragraph is not completed and not clear for the non-expert, average expert, or new researcher. It should rewrite with a clear conclusion.
Reply: We have carefully considered your comments, supplemented and polished the INTRODUCTION- First paragraph. All modifications are highlighted.
At the same time, it also has some shortcomings, including the need for large-scale instruments, inconvenience, poor long-term sensitivity, and physiological indicators that often need to be tested in vitro. In addition, Heubner et al.[5] wrote that there were some shortcomings in the use of a 2-electrode half-cell system for the test of electrochemical performance (such as the continuous decline in the potential stability of the working electrode and cross-talk phenomena). Thus it is often used with other techniques such as liquid chromatography, microneedle, and photochemistry.
- Figure -1 should be redrawn using for a better looking.
Reply: We have carefully considered your comments and redrawn figure-1.
- The manuscript is written poorly, very hard to understand many sentences. There is no connection between two consecutive sentences. Overall messages are unclear almost in all the paragraphs. Therefore, it should be extensively checked by the native language editor.
Reply: We have carefully considered your comments and we have revised the manuscript from beginning to end, and polished it, all of which are highlighted. We also asked the American Journal Experts to polish our manuscript, corresponding editing certification was attached below.
Thank you again for your valuable advices for our manuscript. They are very helpful to our current and future work.
Thank you very much again for your support of our manuscript.
Reviewer 2 Report
The current manuscript entitled “Electrochemical Microneedles: Sharp Instruments in Health Care” by Prof. Gao and Co-workers deliberated on an interesting and emerging scientific topic. The authors detailly deliberated on the Electrochemical Microneedles. Specifically, summarized on the electrochemical microneedles in the past two years and the main challenges faced by electrochemical microneedles and the future development directions were prospected. The review written well. The manuscript can be accepted after addressing the following comments.
Comments
1. Revise the introduction section first paragraph, provide some potential information.
2. Provide the appropriate references for the introduction section starting paragraph.
3. Check the abbreviations used in the manuscript. Somewhere the abbreviations are repeating.
4. Scheme 1 is not informative enough.
5. Please be sure with the copyrights (Figures copied from the other figures).
6. Among the various preparation strategies, which strategy is more advantageous.
7. At many places small letters were used in the starting. Specifically, section 4 captions.
8. Thoroughly check the manuscript for grammatical errors.
9. Conclusion should be written as conclusions
10. Provide challenges that are currently facing with the Electrochemical Microneedles.
Author Response
Response to Reviewer Comments
Reviewer comments in normal type
Author response in italics
Verbatim quotes from the revised text are in boldface
Reviewer 2:
The current manuscript entitled “Electrochemical Microneedles: Sharp Instruments in Health Care” by Prof. Gao and Co-workers deliberated on an interesting and emerging scientific topic. The authors detailly deliberated on the Electrochemical Microneedles. Specifically, summarized on the electrochemical microneedles in the past two years and the main challenges faced by electrochemical microneedles and the future development directions were prospected. The review written well. The manuscript can be accepted after addressing the following comments.
Reply: Thank you very much for your approval of our manuscript.
- Revise the introduction section first paragraph, provide some potential information.
Reply: We have carefully considered your valuable comments, and we have revised and polished the first paragraph of the introduction as required. All modifications are highlighted.
Electrochemistry is a science that studies the charged interface phenomenon formed by two kinds of conductors and its changes[1]. Due to its high sensitivity[2], ease of use, rapid response, economy and mobile advantages, it is widely used in wearable electronic equipment[3], clinical application of activated coating time (ACT)[4], sewage treatment, etc. At the same time, it also has some shortcomings, including the need for large-scale instruments, inconvenience, poor long-term sensitivity, and physiological indicators that often need to be tested in vitro. In addition, Heubner et al.[5] wrote that there were some shortcomings in the use of a 2-electrode half-cell system for the test of electrochemical performance (such as the continuous decline in the potential stability of the working electrode and cross-talk phenomena). Thus it is often used with other techniques such as liquid chromatography, microneedle, and photochemistry.
- Provide the appropriate references for the introduction section starting paragraph.
Reply: We have carefully considered the comments of the reviewers, and we have revised and polished the introduction section starting paragraph as required. All modifications are highlighted.
Electrochemistry is a science that studies the charged interface phenomenon formed by two kinds of conductors and its changes[1]. Due to its high sensitivity[2], ease of use, rapid response, economy and mobile advantages, it is widely used in wearable electronic equipment[3], clinical application of activated coating time (ACT)[4], sewage treatment, etc. At the same time, it also has some shortcomings, including the need for large-scale instruments, inconvenience, poor long-term sensitivity, and physiological indicators that often need to be tested in vitro. In addition, Heubner et al.[5] wrote that there were some shortcomings in the use of a 2-electrode half-cell system for the test of electrochemical performance (such as the continuous decline in the potential stability of the working electrode and cross-talk phenomena). Thus it is often used with other techniques such as liquid chromatography, microneedle, and photochemistry.
- Check the abbreviations used in the manuscript. Somewhere the abbreviations are repeating.
Reply: Thank you for your patient review. We carefully considered the comments of the reviewers and revised all the repeated abbreviations. All the revised parts have been highlighted.
microelectromechanical system (MEMS)
reference electrode (RE)
magnetic resonance (MR)
- Scheme 1 is not informative enough.
Reply: Thanks to your patient review, we have fully considered your comments, and have redrawn Scheme 1.
- Please be sure with the copyrights (Figures copied from the other figures).
Reply: Thanks to the reviewers for their patient review, we have ensured that all copyrights have been obtained, and all modifications have been highlighted.
- Among the various preparation strategies, which strategy is more advantageous.
Reply: Thanks to the reviewers for their patient review. We fully considered the reviewers' comments and compared the advantages of the preparation strategies.
Each preparation strategy has its own advantages and disadvantages. The manufacturing cost of 3D printing technology and magnetorheological stretching photoetching technology is low, and the operation method is simple. In addition, the latter does not need masks and light irradiation and simultaneously does not need to adjust the stretching temperature. However, the resolution of the microneedle electrodes prepared by the two methods is lower. The two-step soft lithography technology and the two-photon polymerization technology can provide high resolution and repeatability but have high manufacturing cost and complex steps, and the maximum manufacturability height of the microneedle array prepared by the two-step soft lithography technology is limited.
- At many places small letters were used in the starting. Specifically, section 4 captions.
Reply: Thank you for your patient review. We carefully considered the comments of the reviewers and revised the manuscript. All the revised parts have been highlighted.
- Drug release
4.1. Release of insulin
4.2. Drug release to bone tissue
4.3. Delivery of adriamycin
4.4. Delivery of glucocorticoids
4.5. Summary
- Thoroughly check the manuscript for grammatical errors.
Reply: Thank you for your patient review. We carefully considered the comments of the reviewers, revised the grammatical errors of the manuscript and polished it, and all the revised parts were highlighted. We also asked the American Journal Experts to polish our manuscript, corresponding editing certification was attached below.
- Conclusion should be written as conclusions
Reply: Thanks to your patient review. We have made changes as required and all changes are highlighted.
- Conclusions and Perspectives
- Provide challenges that are currently facing with the Electrochemical Microneedles.
Reply: Thanks for your patient review, we have fully considered your comments, provided some challenges that electrochemical microneedles are currently facing, and polished them, and all the modifications have been highlighted.
Although electrochemical microneedle technology has developed rapidly over the years, there are still some limitations and challenges, including the fact that microneedles are easily broken due to their large dimensions, inability to sufficiently release a drug, difficulty in stably inserting the skin for a long time, manufacturing costs that are too high to be promoted, materials that have insufficient biocompatibility, drug loading that is too small, and insufficient mechanical strength. In addition, its repeatability and stability will be interfered with by the outside world due to its high sensitivity. Therefore, there is still a long way to go to address these challenges.
Thank you again for your valuable advices for our manuscript. They are very helpful to our current and future work.
Thank you very much again for your support of our manuscript.
Reviewer 3 Report
Overall comments:
The article provides a concise review on development and application of electrochemical microneedle technology in pharmaceutical and medical applications in last two years.
The article provides information on different manufacturing designs, technologies, and materials used for microneedle fabrication along with respective potential application domains. Such information will be highly useful for researchers actively working in area of microneedle technology.
The article can be accepted after minor revisions and taking care of following comments.
Comments
1. Figure 2 is too crowded making text illegible to read and understand. appropriate revision is recommended.
2. Line 86; Author states "major powers promotes".
Who are major powers? Use of such statements should be avoided.
3. Section 2 "monitoring of physiological indicators" and Section 4 "Drug release" comes under applications. However, Section 3 "Preparation ....." focuses on fabrication. The flow of text does not read well, and section 3 now looks misplaced.
A Rearrangement of sections is recommended to enhance readability and flow of the text.
4. Authors overlooked highlighting the limitations and challenges in application of microneedle technology. There are challenges such as size, dimension, fabrication, stability, breakage of needles, ease of use etc. Few of these limitations must be mentioned to provide a complete and comprehensive state of the art of the domain.
5. The writing style is appropriate. However, article should be checked for few typos and grammatical errors present is article.
Author Response
Response to Reviewer Comments
Reviewer comments in normal type
Author response in italics
Verbatim quotes from the revised text are in boldface
Reviewer 3:
The article provides a concise review on development and application of electrochemical microneedle technology in pharmaceutical and medical applications in last two years.The article provides information on different manufacturing designs, technologies, and materials used for microneedle fabrication along with respective potential application domains. Such information will be highly useful for researchers actively working in area of microneedle technology. The article can be accepted after minor revisions and taking care of following comments.
Reply: Thank you very much for your approval of our manuscript.
Comments
- Figure 2 is too crowded making text illegible to read and understand. appropriate revision is recommended.
Reply: Thank you for your patient review. We made a modification to Figure 2, taking full account of the reviewers' comments.
- Line 86; Author states "major powers promotes". Who are major powers? Use of such statements should be avoided.
Reply: Thank you for your patient review. We have fully considered the comments of the reviewers and revised the manuscript as required. Similar errors have also been revised, and all the revised parts have been highlighted.
- Section 2 "monitoring of physiological indicators" and Section 4 "Drug release" comes under applications. However, Section 3 "Preparation ....." focuses on fabrication. The flow of text does not read well, and section 3 now looks misplaced. A Rearrangement of sections is recommended to enhance readability and flow of the text.
Reply: Thank you for your patient review. We have fully considered the comments of the reviewers and made changes to the manuscript section order as required. All the changes are highlighted.
- preparation of microneedles and microneedle electrodes
- Monitoring physiological indicators
- Authors overlooked highlighting the limitations and challenges in application of microneedle technology. There are challenges such as size, dimension, fabrication, stability, breakage of needles, ease of use etc. Few of these limitations must be mentioned to provide a complete and comprehensive state of the art of the domain.
Reply: Thank you for your patient review. We have fully considered the comments of the reviewers and made changes to the manuscript content as required. All the changes are highlighted.
Although electrochemical microneedle technology has developed rapidly over the years, there are still some limitations and challenges, including the fact that microneedles are easily broken due to their large dimensions, inability to sufficiently release a drug, difficulty in stably inserting the skin for a long time, manufacturing costs that are too high to be promoted, materials that have insufficient biocompatibility, drug loading that is too small, and insufficient mechanical strength. In addition, its repeatability and stability will be interfered with by the outside world due to its high sensitivity. Therefore, there is still a long way to go to address these challenges.
- The writing style is appropriate. However, article should be checked for few typos and grammatical errors present is article.
Reply: Thank you for your patient review. We carefully considered the comments of the reviewers, revised the grammatical errors of the manuscript and polished it, and all the revised parts were highlighted. We also asked the American Journal Experts to polish our manuscript, corresponding editing certification was attached below.
Thank you again for your valuable advices for our manuscript. They are very helpful to our current and future work.
Thank you very much again for your support of our manuscript.
Round 2
Reviewer 1 Report
It can be accepted in this format